# Spectroscopic Imaging of Sub-Kilometer Spatial Structure in Lower Tropospheric Water Vapor

David R. Thompson[1], Brian H. Kahn[1], Philip G. Brodrick[1], Matthew D. Lebsock[1], Mark Richardson[1], and Robert O. Green[1]

[1]Jet Propulsion Laboratory, California Institute of Technology, Pasadena, CA 91103 USA

**Correspondence:** David R. Thompson (david.r.thompson@jpl.nasa.gov)

**Abstract.** The subgrid spatial variability of water vapor is an important geophysical parameter for modeling tropical convention and cloud processes in atmospheric models. This study maps sub-kilometer spatial structures in total atmospheric column water vapor with Visible to Shortwave Infrared (VSWIR) imaging spectroscopy. We describe our inversion approach and validate its accuracy with coincident measurements by airborne imaging spectrometers and the AERONET ground-based observation network. Next, data from NASA's AVIRIS-NG spectrometer enables the highest resolution measurement to date of water vapor's spatial variability and scaling properties. We find second order structure function scaling exponents consistent with prior studies of convective atmospheres. Airborne lidar data show that this total column measurement provides information about variability in the lower troposphere. We conclude by discussing the implications of these measurements and paths toward future campaigns to build upon these results.

## 1 Introduction

The complex spatial distribution of atmospheric water vapor surrounding clouds and precipitation structures has important consequences for parameterizing moist processes in atmospheric models. At the scale of General Circulation Models (GCMs), water vapor plays an important role in tropical moist convection and its associated precipitation (Tompkins, 2001; Bretherton et al., 2004). The mean and variability of precipitation rate in the tropics are strongly dependent on the atmospheric water vapor (Peters and Neelin, 2006; Holloway and Neelin, 2010), a fact which has implications for parameterizing convection. Another ubiquitous property of convection is its tendency to aggregate (Bretherton et al., 2005). There is evidence the degree of aggregation will change as the climate warms, potentially changing the cloud feedback (Wing, 2019). Models (Muller and Bony, 2015) and observations (Lebsock et al., 2017) suggest that the tendency of convection to aggregate depends on the degree of spatial variance in the water vapor field. Over land surfaces with heterogeneous surface conditions the variability in atmospheric water vapor can be larger and is seen as a critical component of the timing of deep convection (Stirling and Petch, 2004; Wulfmeyer et al., 2006). These variations in water vapor over convective continental environments are primarily driven

by variability below 2 km altitude and within the Planetary Boundary Layer (PBL) (Couvreux et al., 2009). Accurate water vapor parameterization is also important for Cloud-Resolving or Convection-Permitting models operating at kilometer scales, and Large Eddy Simulations at sub-kilometer resolution. Across all scales, water vapor variability, and its coupling to cloud types and multi-scale organization, is key for advancing the parameterization and simulation of cloud processes.

To the end of advancing remote observations of atmospheric water vapor, this paper focuses on a specific measurement that is independently useful and also typifies the more general challenge of observing variability. *Structure functions* measure the change in the vapor field as a function of distance, quantifying its spatial texture across scales. They are used to analyze many atmospheric components including temperature, winds, and trace gas concentrations, at scales from tens to hundreds of kilometers (Nastrom et al., 1986; Cho et al., 1999). water vapor structure functions can indicate the recent history of the air mass. They can distinguish convective and non-convective systems (Selz et al., 2017), and indicate precipitation rates embedded within a moist column of air with variable levels of column water vapor (Edwards et al., 2019). For a one-dimensional field $f(i)$ indexed by location $i$, the $n$th order structure function $S_n(r)$ is defined as:

$$S_n(r) = E\big[\,|f(i+r) - f(i)|^n\,\big] \tag{1}$$

where $r$ is a separation distance between pairs of points, and $E$ is the expectation over locations. More generally, $S_n(r)$ can represent variability along one direction of a multidimensional field. At least one study of water vapor data has found minor differences in satellite cross-track and along-track directions (Pressel and Collins, 2012). However, it is more typical to assume that water vapor scaling has no preferred horiontal orientation, and that the structure function is rotationally symmetric. $S_n(r)$ is estimated using the mean of observed water vapor values at different spatial offsets. Over a restricted range of distances, structure functions can be described with a power law:

$$S_n(r) \propto r^{\zeta_n} \tag{2}$$

where $\zeta_n$ is the *scaling exponent* of order $n$. The scaling exponent of order two is related to the commonly-used Fourier power spectrum exponent $\beta$:

$$\beta = -(\zeta_2 + 1) \tag{3}$$

These values can diagnose specific atmospheric transport processes. Following Kolmogorov theory, a passive tracer in turbulence has a theoretical second-order scaling exponent $\zeta_2$ of 2/3, or equivalently, a Fourier power spectrum exponent $\beta$ of -5/3 (Pope, 2001).

Previous studies have used a range of instruments to measure water vapor structure functions. In situ aircraft sensors measure a one-dimensional time series along the aircraft flight track. Nastrom et al. (1986) measured separation distances from 150 to 1500 km using this technique. They found $\zeta_2$ ranging from the theoretical value of 2/3 to unity. Later Cho et al. (1999) found $\zeta_2$ consistent with 2/3 over distances of several kilometers. These surveys have recently been augmented by airborne LiDAR measurements, which provide vertical profiles along the flight track. A series of LiDAR observations by Fischer et al. (2012, 2013) measured airmasses at scales down to 2 km. They reported $\zeta_2 = 0.6 - 0.75$ in convective airmasses, and $\zeta_2 = 1.0 - 1.2$ in

non-convective airmasses. Convective environments had shallower scaling indicating higher spatial variability at small scales. They hypothesized that this flatter power spectrum slope, which was also common in boundary layer airmasses, was related to the small-scale injection of water vapor anomalies by convective eddies. However, their study did not conclusively identify the cause. More recently, Selz et al. (2017) compared airborne LiDAR measurements with simulations at scales greater than 11 km. This study confirmed that power law exponents were strongly related to altitude and the presence or absence of convection. They showed $\zeta_2$ above 1 for nonconvective airmasses in the free troposphere, near 0.6 in the boundary layer, and as low as 0.2 in convective airmasses.

Contrasting with localized, high resolution aircraft data, orbital data such as the Atmospheric Infrared Sounder (AIRS) on EOS Aqua have provided more comprehensive power law exponents across the mesoscale and synoptic scales, but without comparing them to GCMs. Kahn and Teixeira (2009) developed a global climatology of scaling properties between 150-1200 km based on vertically-resolved temperature (T) and specific humidity (q) observations by AIRS. Kahn et al. (2011) extended these results and compared them to climate GCMs, MERRA reanalysis, and VOCALS-REx in situ observations of vertically-resolved T and q within and above shallow cumulus off of the coast of Peru. They found strong evidence of $\zeta_2 = 2/3$ scaling in the boundary layer and tropopause at all latitudes, with steeper scaling in the mid-troposphere and at low latitudes. This was consistent with the view of water vapor as a passive scalar in turbulent flow, and implied more small-scale variability than was predicted by contemporaneous GCMs. This increased scaling of the tropical free troposphere was also consistent with theoretical predictions of steeper scaling of column water vapor within and near strongly precipitating convection (Edwards et al., 2019). Interestingly, their analysis of aircraft data from the VOCALS-REx experiment suggested a steepening of the curve at the smallest scales below 10 km (Kahn et al., 2011). This result suggested a possible change in the scaling properties at the finest resolutions.

These studies contribute to a growing body of literature on water vapor scaling. However, important gaps remain. High spatial resolution data are sparse, and most studies explore spatial scales above 1 km. To our knowledge no study has yet corroborated the Kahn et al. (2011) steepening phenomenon at scales less than 10 km, or probed the structure functions at scales less than 2 km. Most aircraft data consist of one dimensional time series, rather than the two-dimensional maps available from instruments like AIRS. Such measurements confined to the flight trajectory provide fewer samples, increasing uncertainty in the derived exponents (Selz et al., 2017; Guillaume et al., 2018).

A new generation of orbital instrumentation may shed new light on fine-scale water vapor. Visible to Shortwave Infrared (VSWIR) imaging spectrometers, such as NASA's upcoming EMIT mission (Green et al., 2019) or the expected Surface Biology and Geology (SBG) investigation (National Academies of Sciences and Medicine, 2018), are highly sensitive to the water vapor column absorption (Shivers et al., 2019). They typically have a spectral resolution of 5-10 nm and span the 380 - 2500 nm interval, a range which overlaps significant water absorption features. A typical pushbroom instrument could have as many as 1200 cross-track measurements with ground sampling distances of 30-60 m. Most such investigations target surface properties. However, a byproduct of this analysis will be accurate column-averaged water vapor measurements at high spatial resolution over wide areas. The spatial resolution and accuracy of the derived water vapor column measurements will be unprecedented, providing opportunities to probe the horizontal variability of water vapor on global scales. Before

these missions launch, archives of airborne precursor data provide an opportunity to validate the technique and begin the investigation.

This manuscript demonstrates direct mapping of the sub-kilometer spatial structure in column water vapor using an airborne VSWIR imaging spectrometer. We first describe our model for vapor absorption features in the near- to shortwave infrared. We estimate clear-sky vapor concentrations by inverting a combined model of the atmosphere and surface reflectance after filtering out cloud-affected footprints. We validate the approach with NASA's "Classic" Airborne Visible Infrared Imaging Spectrometer, AVIRIS-C (Green et al., 1998), evaluating overflights of the AERONET observation network (Holben et al., 1998). We analyze imaging spectroscopy data acquired during the "Next Generation" Airborne Visible Infrared Imaging Spectrometer (AVIRIS-NG) India campaign of 2018. This campaign includes several scenes with highly favorable solar geometry, providing a uniquely high resolution measurement of the $H_2O$ vapor column. This enables estimation of the $H_2O$ spatial structure functions at sub-kilometer scales. We find confirming evidence of $\zeta_2 = 2/3$ scaling in some but not all atmospheres, with Kahn et al. (2011) curve steepening that continues down to at least 100 m. Finally, we discuss the relationship between the total column measurement and variability in the lower troposphere. We assess water vapor profiles from airborne LiDAR campaigns (Bedka et al., 2020). These data show that lower tropospheric variability consistently dominates the total column, making the VSWIR measurement informative about lower atmospheric water vapor. We conclude by discussing the implications of these measurements and future campaigns that build upon the results.

## 2  Methods

### 2.1  Atmospheric Model

Our technique estimates water vapor independently for each spatial location by inverting an atmospheric radiative transfer model. We define a state vector $x$ containing all the free parameters in the system. It includes the surface reflectance vector $\rho$, the column water vapor concentration $q_v$, and the aerosol optical depth at 550 nm. Aerosol optical properties were derived from a canonical sulfate aerosol type and validated in prior studies of the India campaign (Thompson et al., 2019). A *forward model* $F(x)$ maps this state onto an observed radiance at the sensor, $L_o$:

$$L_o = F(x) + \epsilon \tag{4}$$

Boldface represents a vector or matrix, e.g. $L_o$ has one element for each spectrometer channel. The random variable $\epsilon$ represents instrument noise, distributed according to a zero-mean Gaussian with covariance $\Sigma_e$. We can decompose $L_o$ into different photon paths, with $\circ$ representing element-wise multiplication:

$$L_o = L_{atm} + t \circ L_{dn} \circ \frac{\rho}{1 - S \circ \rho} + \epsilon \tag{5}$$

Here $L_{atm}$ represents the *path radiance* caused by molecular and particle scattering; these photons never reach the surface. The second term represents all the photons that interact with the surface at least once. $L_{dn}$ is the downwelling illumination at the bottom of the atmosphere. $\rho$ is the spectral surface reflectance. $t$ is the atmospheric transmittance from the surface to the

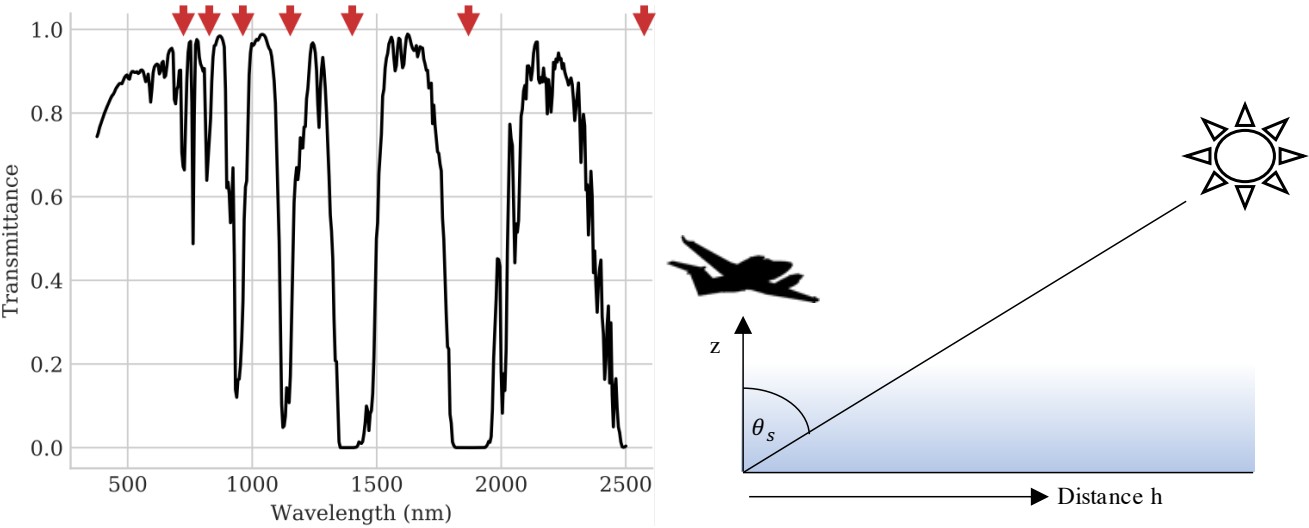

**Figure 1.** Left: Atmospheric transmissivity from 380-2500 nm reveals multiple $H_2O$ absorption features, indicated with red arrows. Right: The spatial sensitivity of solar-reflected measurements depends on water vapor absorption along the optical path from the sun to the ground to the sensor. Aircraft image credit: NASA

sensor along its line of sight. $S$ is the spherical sky albedo observed from the ground. The relation in Equation 5 holds for a locally-homogeneous and Lambertian surface, but small departures from these conditions are not catastrophic and in any case the assumptions hold sufficiently well for the scenes in this study.

Our atmospheric model calculates the optical coefficients $L_{atm}$, $t$, and $S$ using the MODTRAN 6.0 software package (Berk and Hawes, 2017). Specifically, we use the DISORT code with 8-stream multiple scattering calculations. The absorption model uses a correlated-k representation with 0.1 $cm^{-1}$ bins. Vertical profiles are assigned from a 20-layer stratified atmosphere. For computational efficiency, we do not run the complete Radiative Transfer Model (RTM) for each evaluation of $F(x)$. Instead, we calculate optical coefficients in advance to fill a lookup table for each component in the right side of equation 5, and then interpolate within this table to determine the precise radiance for any given state vector. Figure 1 illustrates the atmospheric transmittance in the measurement interval from 380-2500 nm, including several prominent absorption features related to $H_2O$ rovibrational overtones. Absorption features at 940 and 1140 nm carry most of the water column information. Stronger features at 1380 and 1880 nm are saturated at atmospheric path lengths, and consequently less useful.

The remote $H_2O$ measurements aim to quantify the integrated mass of water vapor directly above each pixel in the scene. However, this is not always possible for a solar-reflected signal. Neglecting scattering, which is small in the near infrared, the remote water vapor observation measures absorption along a two-part optical path from the sun to the ground to the sensor. The sun is seldom directly overhead, and its downwelling illumination enters the atmosphere at some horizontal offset from the reflection point (Figure 1, Right). Consequently, a spectrum's spatial sensitivity can extend far beyond the target pixel in the direction of the sun. Projected onto the ground, the vapor absorption path forms a long, thin footprint that extends hundreds

of meters from the target pixel in the sunward direction. To quantify this effect, we define the water vapor spatial sensitivity as the extinction-weighted distance between downwelling and upwelling beams. We use $g(h)$ to represent the relative sensitivity to water vapor at $h$, some horizontal offset distance in the solar direction. For a nadir-pointed observation, assuming that the vertical profile of water vapor is locally constant, the relationship is:

$$g(h) = \begin{cases} \int_{z \in [z_s, z_o]} \kappa(z)\, dz & \text{if } h = 0 \\[2ex] \kappa\left(\frac{h}{\tan \theta_s} + z_s\right) \frac{1}{\cos \theta_s} & \text{if } h > 0 \end{cases} \tag{6}$$

where $\kappa(z)$ is the water vapor number density at altitude $z$. The variables $z_s$ and $z_o$ represent the surface elevation and sensor altitude respectively. The spatial sensitivity is strongly dependent on solar zenith angle ($\theta_s$). It is also dependent to some extent on the view angle; if the observation is off-nadir, $g(h)$ includes the sensitivity of both upward and downward paths. We define the average spatial offset of a measurement, $\mu_h$, as the mean horizontal position of the water vapor along the two-part optical path:

$$\mu_h = \frac{\int_0^\infty h\, g(h)\, dh}{\int_0^\infty g(h)\, dh} \tag{7}$$

In addition to shifting the center of the spatial response, larger solar zenith angles can coarsen the effective spatial resolution by stretching the water vapor sensitivity footprint in the sunward direction. We define the effective spatial resolution, $R_e$, as the symmetric horizontal distance from the center point which encloses 68.2% of the spatial response. This would be equivalent to the area inside a single standard deviation if the response function were Gaussian.

$$0.682 = \frac{\int_{\mu_h - R_e}^{\mu_h + R_e} g(h)\, dh}{\int_0^\infty g(h)\, dh} \tag{8}$$

The left panel of Figure 2 shows the relative sensitivity to water vapor at different horizontal offsets for a nadir-viewing measurement of a tropical atmosphere at 4 km acquisition altitude, similar to the observing geometry of the following experiments. We show the response areas for three solar zenith angles. As the solar angles increase, the response function extends farther from the target pixel, with increasingly "thick tails" caused by water absorption along the slanted downwelling path. The right panel of Figure 2 shows the resulting resolutions as a function of solar zenith angle at different viewing geometries and atmospheric profiles, at sensor altitudes of 4 km and Low Earth Orbit (LEO). It shows that the spatial sensitivity is only weakly dependent on the sensor altitude, and somewhat dependent on the viewing angle and the vertical profile of water vapor in the atmosphere. The solar zenith angle is the dominant influence on spatial resolution, motivating a careful selection of flightlines for the following experiments. The spatial footprint projected on the ground is not radially symmetric; it is long and thin, but retains the native spatial resolution along its short axis. Consequently, for isotropic structure functions the effective spatial resolution is a worst case, and sensitivity improves as one calculates structure functions in directions orthogonal to the sun.

## 2.2 Inversion methodology

To estimate $q_v$, we invert a combined model of surface and atmosphere. We use a Bayesian Maximum A Posteriori formalism (Rodgers, 2000) that is common among atmospheric sounding missions. Recent work extended this approach to the Visible-

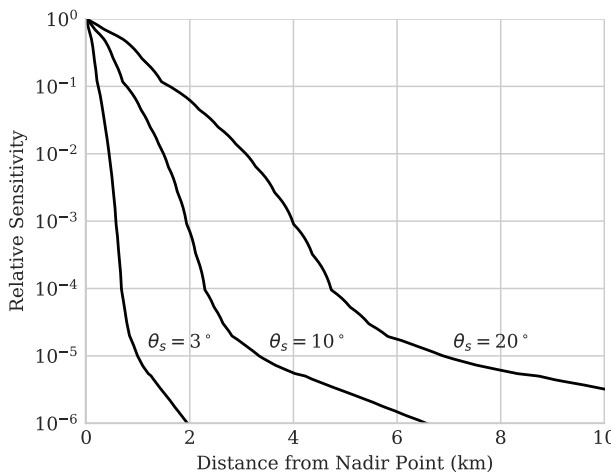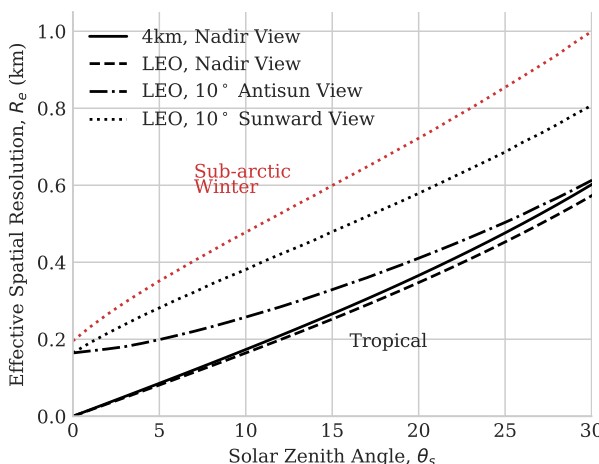

**Figure 2.** Left: the relative sensitivity to water vapor at 4 km acquisition altitude in a tropical atmosphere as a function of horizontal distance from nadir, partitioned by solar zenith angle $\theta_s$. Right: the effective spatial resolution as a function of solar zenith angle $\theta_s$, at both 4 km and low earth orbit (LEO) acquisition altitudes. The red line shows a sub-arctic winter atmospheric profile indicating the range of spatial resolution across two very different atmospheric conditions. In reality, the sun never reaches low zenith angles in polar regions of the globe.

Shortwave Infrared spectral interval (Thompson et al., 2018b, 2019). The solar-reflected regime is strongly influenced by variability in surface reflectance. Consequently, we fit atmospheric parameters simultaneously with a flexible surface model. As described in Section 2.1, our state vector $x$ includes surface reflectance in every channel, the column water vapor concentration $q_v$, and the aerosol optical depth at 550 nm. A forward model $F(x)$ transforms this state vector into a simulated radiance at the sensor following Equation 5.

Our inversion determines the most likely state vector to explain the observation $L_o$. It includes background knowledge with a multivariate Gaussian prior over state vector elements, with mean $\mu_a$ and covariance $\Sigma_a$. We fit this distribution as in (Thompson et al., 2019), with a library of diverse reflectance spectra. Shrinkage regularization (Theiler, 2012) ensures that the inversion can represent spectra not spanned by original library subspace. The inversion balances this prior against the measurement noise in Equation 5, which is determined from a component-wise instrument performance model (Thompson

et al., 2020). The optimal state vector minimizes the following cost function:

$$\chi(x) = (F(x) - L_o)^T \Sigma_e^{-1} (F(x) - L_o) + (x - \mu_a)^T \Sigma_a^{-1} (x - \mu_a) \tag{9}$$

where $\Sigma_e$ is the instrument noise. This cost is proportional to the negative logarithm of the posterior probability, a product of multivariate Gaussian prior and likelihood terms. We solve it with a trust region gradient descent optimization, a nonlinear optimization approach that respects positivity constraints on the free parameters (Lenders et al., 2018) In principle, any gradient

based optimization would suffice with an appropriate starting point. The inversion typically converges in 10-20 iterations.

## 2.3 Postprocessing and structure functions

Before analyzing the spatial structure of the resulting $H_2O$ maps, we perform several postprocessing steps to improve the map consistency. Even a highly accurate retrieval is likely to suffer some biases due to the influence of the surface type or magnitude. For example, very slight errors in atmospheric path radiance estimates are proportionally larger for dark targets, which can induce a spurious dependence between albedo and water vapor. Surface reflectance features that overlap water vapor absorptions can also influence retrievals. Finally, minor differences in the radiometric response or linearity of different cross-track elements can create striping artifacts in derived products. We address all of these issues with a single empirical correction. We first define a grid of locations $i \in \mathcal{L}$ where the retrieval is performed. We assert that the retrieved water vapor at each location $i$, written $\hat{q}_i$, is the combination of an underlying "true" water vapor signal $q_i$, zero-mean measurement noise $\epsilon_q$, and non-stochastic interference by surface and systematic instrument effects. We define a feature vector $\psi_i$ to include the surface reflectance in each channel, and a sparse position vector $\boldsymbol{p}$ encoding the associated cross-track position on the Focal Plane Array (FPA) in a stacked binary representation. The position encoding enables the model to represent radiometric sensitivity of different FPA elements. The column vector $\psi_i$ thus combines $\rho_i$ and $\boldsymbol{p}_i$. For simplicity, we assume the surface interference is a linear combination of these features, weighted by coefficients $\phi$:

$$\hat{q}_i = q_i + \boldsymbol{\phi}^T \boldsymbol{\psi}_i + \epsilon_q \tag{10}$$

To estimate $\phi$, we treat the true water vapor component as a random variable. Over large spatial scales, biases due to diverse content will tend to average out, and the sample mean of retrieved values can be used to approximate the true mean. Rearranging algebraically, we define the local vapor anomaly $\mathcal{A}_i$ as:

$$\mathcal{A}_i = \hat{q}_i - \left[ \frac{1}{|\mathcal{L}|} \sum_{j \in \mathcal{L}} \hat{q}_j \right] = \boldsymbol{\phi}^T \boldsymbol{\psi}_i + \epsilon_w \tag{11}$$

Where $\epsilon_w$ now incorporates variability due to retrieval noise as well as variability in the true water vapor field. Since $\epsilon_w$ is zero mean, Equation 11 reduces to a straightforward linear regression problem. For a data matrix $\boldsymbol{U}$ with rows made up of all reflectance spectra, and a vector $\mathcal{A}$ of anomaly values, we estimate $\phi$ with ordinary least squares regression. We subtract the predicted error from the original estimate to yield the bias-corrected estimate $q_i^{'}$:

$$q_i' = \hat{q}_i - \hat{\boldsymbol{\phi}}^T \boldsymbol{\psi}_i \quad \text{for} \quad \hat{\boldsymbol{\phi}} = \boldsymbol{U}(\boldsymbol{U}^T\boldsymbol{U})^{-1}\boldsymbol{U}^T\mathcal{A} \tag{12}$$

To apply the bias correction, we segment long flightlines into segments of no longer than 2000 pixels each, and apply the interference correction to each segment independently.

A final postprocessing operation smooths each water vapor image with a Gaussian spatial filter. This dramatically reduces the retrieval noise, making it possible to resolve much finer structures. This noise reduction is also beneficial for estimating structure functions, which are bounded artificially on the low end by the average squared noise in neighboring pixels. We use leave-one-out cross-validation (Shao, 1993) to select an optimal blurring kernel width. To score a candidate width, we compare every point in the scene to the prediction made by applying the blurring kernel to that location, excluding the point under test

with appropriate renormalization. The optimal kernel standard deviations range from 4 to 6 in different scenes; we use the lower value to preserve fine spatial structure at sub-100m scales.

After these postprocessing steps, we calculate empirical structure functions for each pixel shift distance d, computing the squared differences between image locations shifted in the along-track direction. Recall that the long flightlines are corrected in parts no longer than 2000 pixels in length, revealing structure functions up to 4 km. To avoid minor offsets in the vapor field on each side of a border, we do not permit shifts across multiple segments. We aggregate the statistics of all segments to form a structure function estimate for each flightline. We manually mask any visible artifacts in water vapor images to exclude them from the calculation. These artifacts are primarily due to the presence of clouds or their shadows. Clouds can also disrupt their local light fields with scattered illumination, an effect visible in water vapor maps as obvious halos around clouds and their shadows. To mitigate this effect, we dilate the masks by 200 m in all directions.

## 3   Results: AERONET Overflights

We first assess the absolute accuracy of water vapor absorption measurements using airborne overflights of the AERONET robotic observation network (Holben et al., 1998). The AERONET sunphotometers view the sun directly, estimating water vapor by solar extinction (Pérez-Ramírez et al., 2014). Consequently, they do not measure the same optical path as the downlooking sensor. Nevertheless, these coincident overflights validate the approach and provide a ceiling for unmodeled uncertainties. The airborne instrument is AVIRIS-C (Green et al., 1998). It flew on an ER-2 aircraft at approximately 20 km altitude. Our dataset spans six years of operations over California, from 2013-2019, during which it overflew active terrestrial AERONET sites on over 100 occasions. Our atmospheric model for these observations used an Air Force Geophysics Laboratory (AFGL) midlatitude summer profile (Anderson et al., 1986). Aerosols were light throughout the overflights, so we omitted them from the state vector.

We use several filtering methods to remove uncertain datapoints. Many overflights are contaminated by clouds. Our comparison excludes obvious clouds recognized by an estimated surface reflectance at 450 nm above 0.2. However, even when opaque clouds do not directly cover the aeronet location, high-altitude cirrus, cloud shadows or scattered irradiance can contaminate a retrieval. Additionally, high spatiotemporal variability in water vapor can cause a discrepancy; it would aggravate the difference in optical paths, as well as the imperfect temporal coincidence between the AERONET acquisitions and the overflight. Considering the strong influence of $\theta_s$ demonstrated in Figure 2, and that $\theta_s$ during typical flights often reaches 45 degrees or greater, the ground-based measurement could see a very different vapor field if humidity were not horizontally homogeneous. To address these issues, we remove any datapoints where the in-situ standard deviation in $H_2O$ is larger than 0.1 g cm$^{-2}$, estimated by comparing the 10 temporally closest acquisitions. After this filtering step, 64 datapoints remain. Some discrepancies in the optical paths remain, which become larger for column water vapor in the free troposphere than in the planetary boundary layer. As neither measurement resolves the detailed vertical profile, it is not possible to distinguish these cases without additional information.

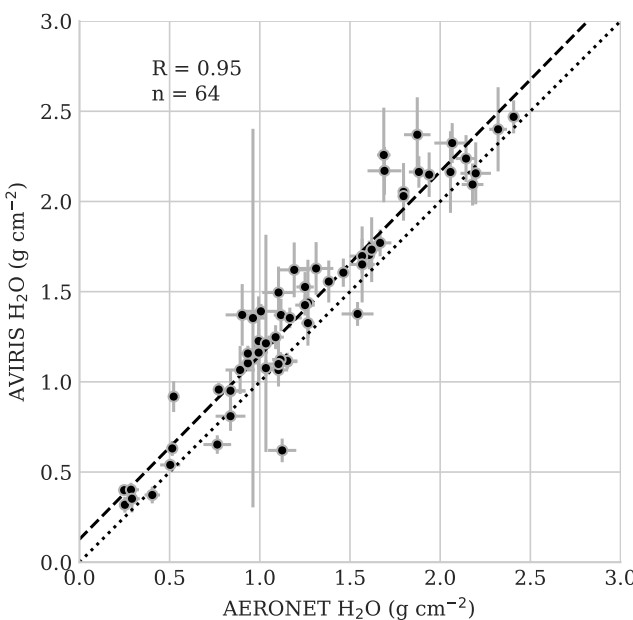

**Figure 3.** Coincident AVIRIS-C vs AERONET water vapor measurements, filtering out AERONET instances with high temporal variability ($1\sigma > 0.1$ g cm$^{-2}$ over 10 temporally closest measurements).

Figure 3 compares the AERONET and remote retrievals for all sites. Error bars indicate the variability in water vapor over the 10 closest timesteps (AERONET) or the 10x10 enclosing rectangle of pixels (AVIRIS-C). Each AERONET timestep is approximately 5 minutes, so our window provides a measure of variability in the hour around the flight. These overflights represent a wide range of atmospheric conditions and solar angles. They span column water vapor concentrations from approximately 0.3 to 2.5. The airborne and uplooking measurements show strong agreement, with a correlation coefficient $R$ of 0.95. A few AVIRIS-C observations with very high variability are likely clouds or cloud shadows that survived the filtering process. Nevertheless, the result is broadly consistent with AERONET accuracy of 12–15% for column water vapor, a claim which has been validated independently by comparison to microwave radiometers (Pérez-Ramírez et al., 2014). This provides confidence in the accuracy of the water vapor retrieval. There is a small bias between the two datasets. However, even if we attributed it entirely to the airborne data such an offset would not influence the structure function measurement.

## 4   Results: Structure Function Measurement

We apply the H$_2$O retrieval methodology to four flightlines from a 2018 AVIRIS-NG campaign in India. We selected these flightlines for favorable solar observing geometry and because they represented two distinct conditions observed on different

days. All flightlines were at 4 km altitude over water, providing a uniform, topographically-flat surface for vapor retrievals. On May 12, two flightlines encountered scattered low clouds. On May 14, two more flightlines encountered clear skies. We restricted our study to flights with solar zenith angles less than 10 degrees, a condition which occurred on two flight days. This provided effective spatial resolutions from 80 to 250 m at nadir. In comparison, the native spatial sampling of AVIRIS-NG at these altitudes was 4 m. In addition to improving spatial resolution, the extreme solar angles also produced significant sunglint, revealing near infrared water features that would otherwise be masked by the absorption of the water surface. Information on each flightline appears in Table 1. We applied the standard AVIRIS-NG radiance calibration procedure (Chapman et al., 2019), including corrections to the spectral response function (Thompson et al., 2018a). We then performed water vapor retrievals on all flightlines using a tropical atmospheric profile (Anderson et al., 1986). AVIRIS-NG had a higher intrinsic Signal to Noise Ratio (SNR) than AVIRIS Classic, and the solar angles provided more signal than a typical AERONET overflight from the validation experiment. Consequently, we expected a more sensitive retrieval in the India datasets.

To provide context for interpreting the vapor fields, we analyzed MERRA-2 reanalysis data (Gelaro et al., 2017) for each of the two days. The atmospheric conditions were generally similar across the overflights, with light trade winds and the lack of an obvious inversion to stratify boundary layer processes. There were also some differences between the days. Wind velocity changed slightly, but was not obviously tied to any change in atmospheric turbulence. The relative humidity was generally higher on the 14th. The lapse rate was slightly more variable: 8 K km$^{-1}$ at 760 hPa as opposed to 7 K km$^{-1}$ on the 12th. These changes would be consistent with a slightly more turbulent atmosphere and a shallower scaling exponent, though there was no obvious step change in atmospheric stability. Figure 4 shows the temperature, lapse rate, specific and relative humidity for each day.

| Label | Flightline ID | Date | $\theta_s$ | Nadir $R_e$ (m) | Latitude N | Longitude W | Length (m) |
|-------|---------------|------|-----------|-----------------|------------|-------------|------------|
| A | ang20180512t052609 | 12 May 2018 | 9.7 | 250 | 21.649 | 87.775 | 24000 |
| B | ang20180512t053942 | 12 May 2018 | 6.9 | 177 | 21.639 | 87.789 | 24000 |
| C | ang20180514t055115 | 14 May 2018 | 4.1 | 105 | 21.524 | 88.346 | 8000 |
| D | ang20180514t060206 | 14 May 2018 | 3.1 | 79 | 21.526 | 88.325 | 8000 |

**Table 1.** Flightlines used for structure function analysis.

Figure 5 shows a typical spectrum fit. The top panel shows the radiance measurement. The converged model matches the measurement closely, with largest discrepancies in the shortest wavelengths. The middle panel shows the residual error. Some structure at the sub-2% level is consistent with minor model discrepancies or calibration errors. These mostly affect the shortest wavelengths outside the dominant water absorption intervals. The reflectance spectrum in the bottom panel shows a good quality retrieval of a water surface. There is considerable sunglint, a spectrally-flat additive signal that uniformly elevates the spectrum. This raises the near- to shortwave-infrared reflectance, which would normally be zero over liquid water, to values of 12-13%. Elevated reflectance in the presence of sunglint enables high-accuracy retrievals of $q_v$ over open water. This is a reliable consequence of small solar zenith angles under a wide range of wind conditions.

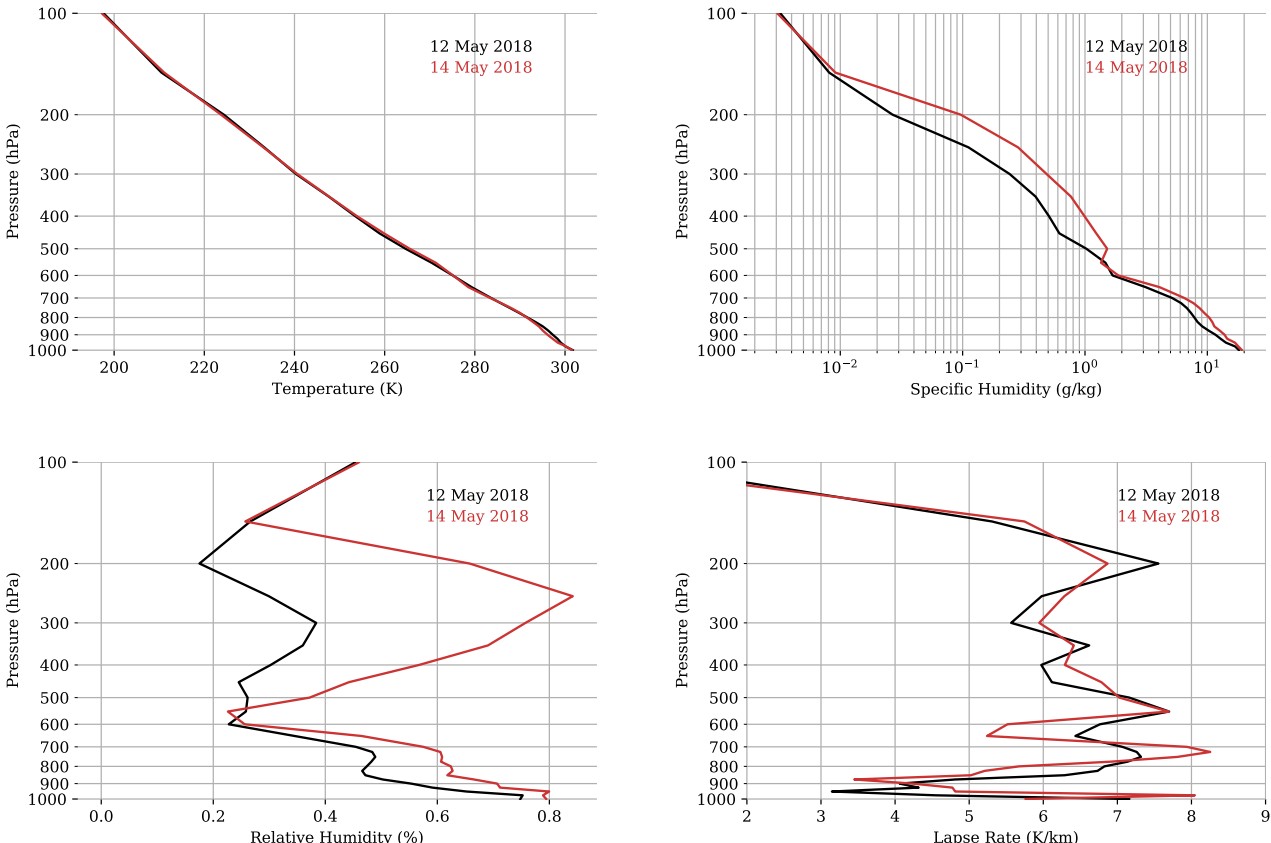

**Figure 4.** MERRA-2 reanalysis predictions for the two flight days. Counterclockwise from upper left: temperature, specific humidity, relative humidity, and lapse rate.

Figure 6 shows the column water vapor maps for individual segments of the flightlines. In the top panels, dark blue areas indicate manual masks applied to exclude cloudy regions. After the interference correction, the vapor maps are generally free of artifacts except for a minor band at the zero phase angle. This artifact, indicated by a white arrow, may be related to issues with the sun glint bidirectional reflectance distribution or the aircraft shadow. Therefore, it was excluded from the statistics.

The smallest-scale variability is almost certainly related to noise in the retrieval algorithm. Larger scale coherent structures show that meaningful differences in water vapor content are resolved at the level of 1-3%, over scales comparable to the predicted spatial resolutions of 80-250 m. The center and bottom panels show example maps for flightlines C and D on May 14, which were entirely free of clouds. In aggregate, the flightlines encountered water vapor values ranging from about 3.23 to 3.42 g cm$^{-2}$. The smoothing operation dramatically increases the contrast of spatial features (Figure 7).

Figure 8 shows structure functions for the four flightlines with second-order scaling exponents for representative intervals. Grey lines indicate the best-fitting power law exponents over the 500-1000m interval. We also plot the canonical 2/3 slope for

reference. The flightlines' configurations of vapor and clouds were all unique, but their scaling functions from the common flight days show similar profiles. This intra-day similarity demonstrates the repeatability of the measurement in a consistent airmass. For flightlines A and B, $\zeta_2$ is $0.64$ and $0.63$ respectively. These values are very close to the classic 2/3 value of a passive scalar in turbulence. The slope continues to scales well under 1 km, with a steepening of the curve below about 500m. This corroborates the VOCALS-REx data analyzed by Kahn et al. (2011), although those observations were taken in a stable regime within and above stratocumulus clouds off of the coast of S. America. In this case, the observed steepening continues to scales approaching the effective spatial resolution of approximately 250 m. The right panel shows flights C and D on May 14. These flightlines have shallower profiles in the 500-1000 m range, with scaling exponents of $0.41$ and $0.29$. The curves are less consistent across the two overflights, which might be due to small sample sizes. Flightlines C and D contained just one segment of 2000 m, compared with six segments in each of flightlines A and B.

To confirm that the results were robust to slight differences in viewing angles, we recalculated the same result using the central half of the flightline data. This excluded all off-nadir view angles greater than approximately 9 degrees. The results did not change significantly; flightlines A,B,C and D showed $\zeta_2$= 0.55, 0.58, 0.41 and 0.29 respectively. These differences would not be large enough to change our interpretation. In all cases, the maps comprise millions of independent measurements making it possible to resolve structures of magnitudes smaller than the single-pixel noise. All acquisitions, including the extreme shallow scaling of flightlines C and D, are in the range of exponents predicted for convective airmasses by Selz et al. (2017). This is consistent with the presence of convective clouds in the flightlines.

## 5   Discussion

The airborne experiments demonstrate the ability of VSWIR spectroscopy to measure total column water vapor with high precision, but cannot directly apportion this observed variability to different vertical layers. The state of the lower troposphere generally dominates the overall water vapor content, but it does not necessarily follow that the troposphere determines the observed variability over short spatial scales. Our derived $\zeta_2$ coefficients refer to the horizontal variability of water vapor integrated along the full solar-reflected optical path, and variability at different altitudes could contribute to estimated $\zeta_2$. For example, one might observe a superposition of different atmospheric regimes at different layers, such as convective and non-convective, or boundary-layer and free-troposphere, which would blur the variability in the total column. This could be significant for attributing the variability or for comparing the observations with more localized measurements. Given that this study is one of the first that explores water vapor variability at such high horizontal resolution, there is a dearth of independent data against which to interpret vertical sensitivity.

We can gain insight by framing the relationship between full column and stratified variability as a statistical question. Other vertically-resolved measurements give evidence that the derived exponents reflect structure in the lower troposphere and planetary boundary layer. Preliminary support comes from reprocessed data from the five 2019 Cal/Val campaign flights of the High Altitude Lidar Observatory (HALO) over the Eastern Pacific (Bedka et al., 2020). The HALO's differential absorption lidar (DIAL) water vapor retrievals were reprocessed for our analysis with 3 km along-track averaging, 0.5 km vertical averaging

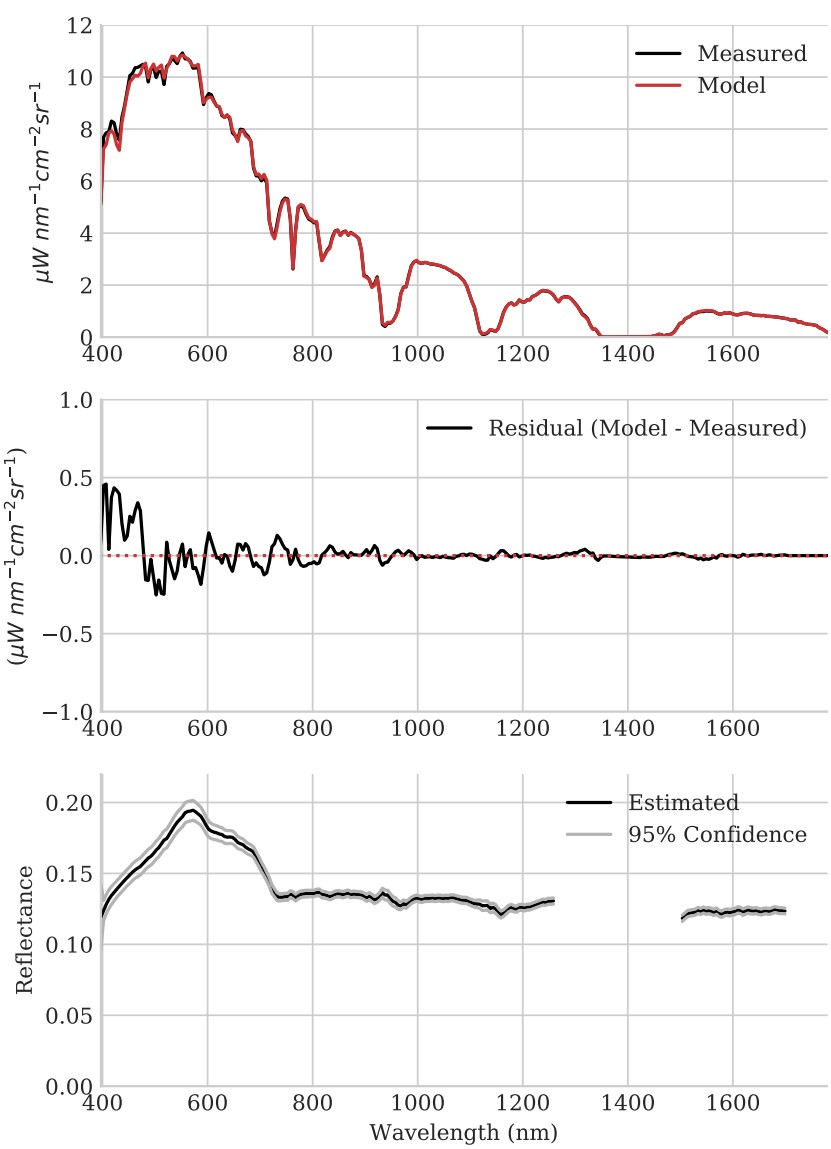

**Figure 5.** Top: Example radiance spectrum. Middle: Model fit residual. Bottom: Estimated reflectance with 95% confidence bounds.

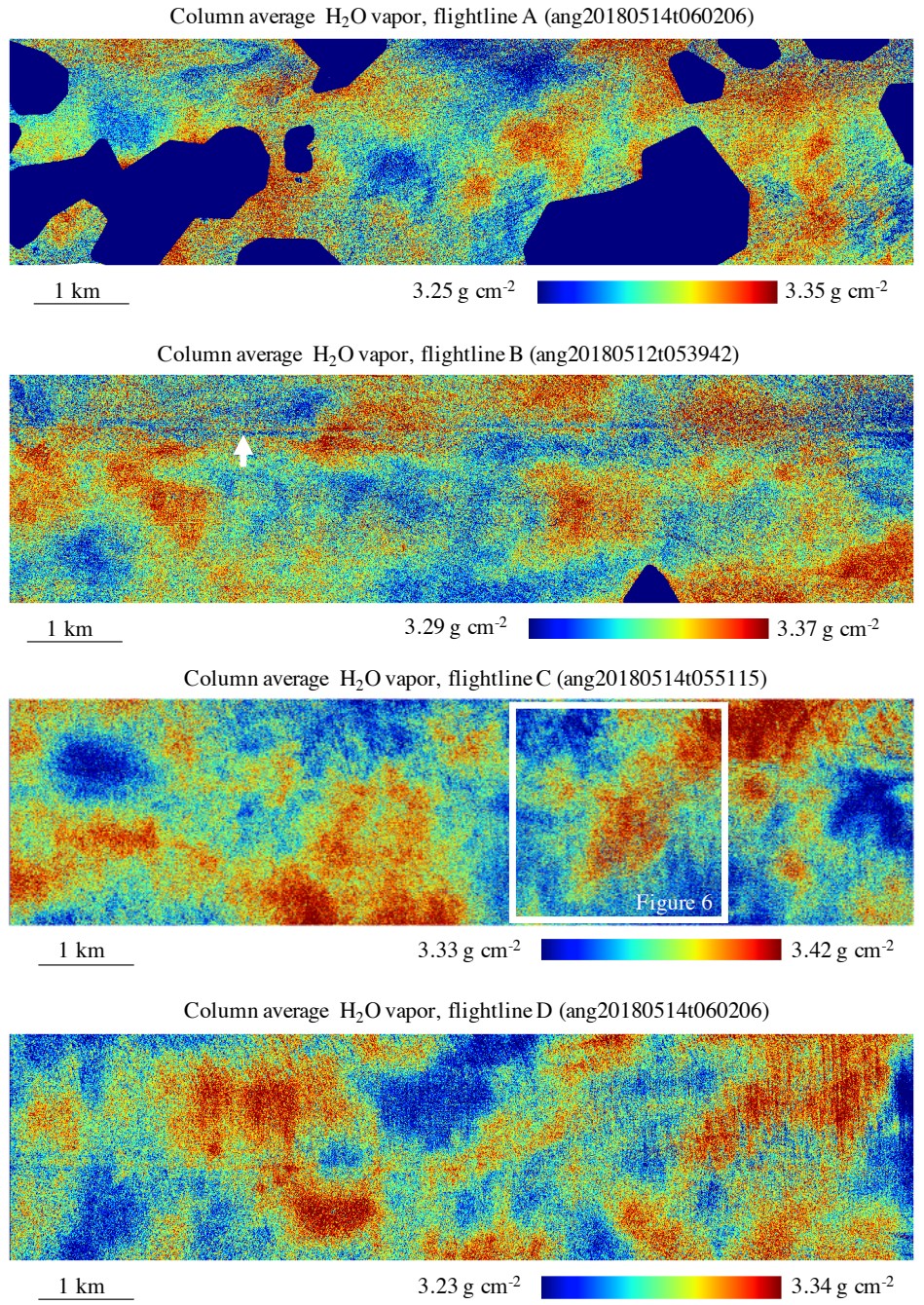

**Figure 6.** H₂O vapor maps. In flightlines A and B, dark blue areas are masked to avoid clouds. A white arrow in flightline B indicates an artifact that occurs where the solar phase angle is zero. A white box in flightline C indicates the area shown by Figure 7.

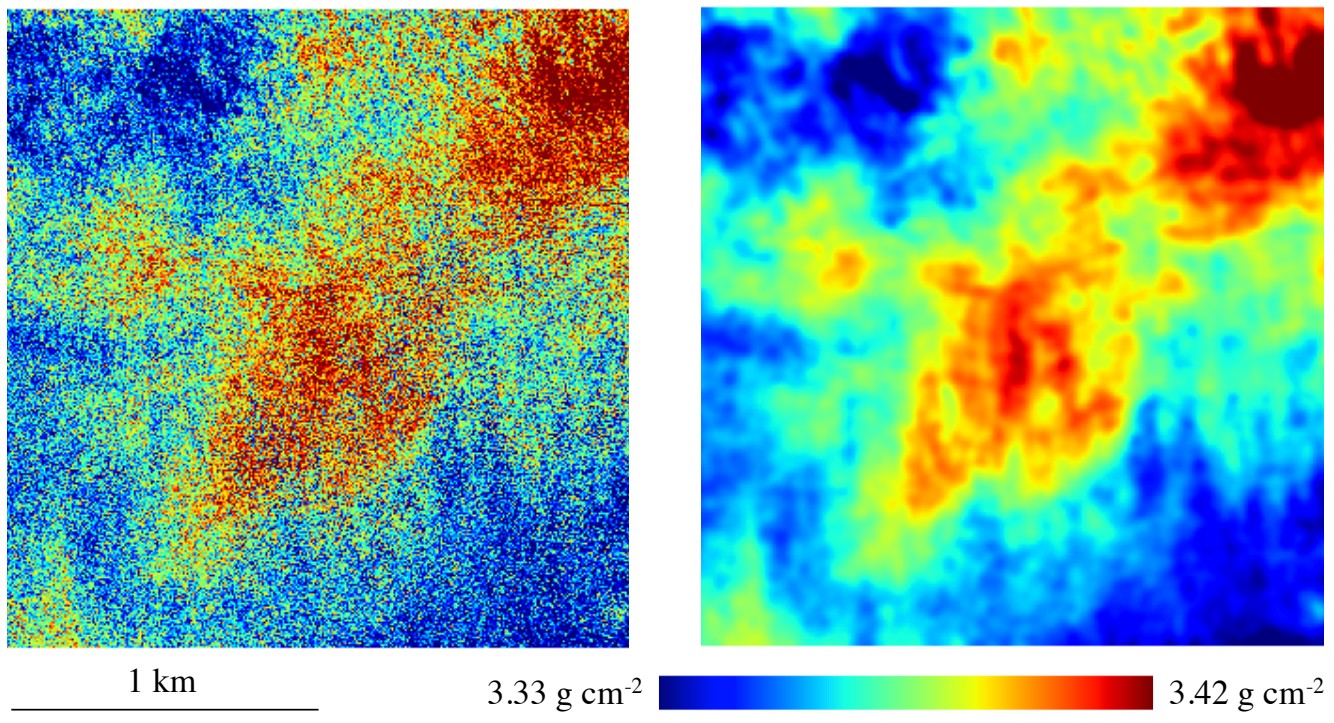

**Figure 7.** Left: Initial water vapor map from flightline C. Right: Noise reduction after kernel smoothing reveals fine-scale structure at sub-kilometer scales.

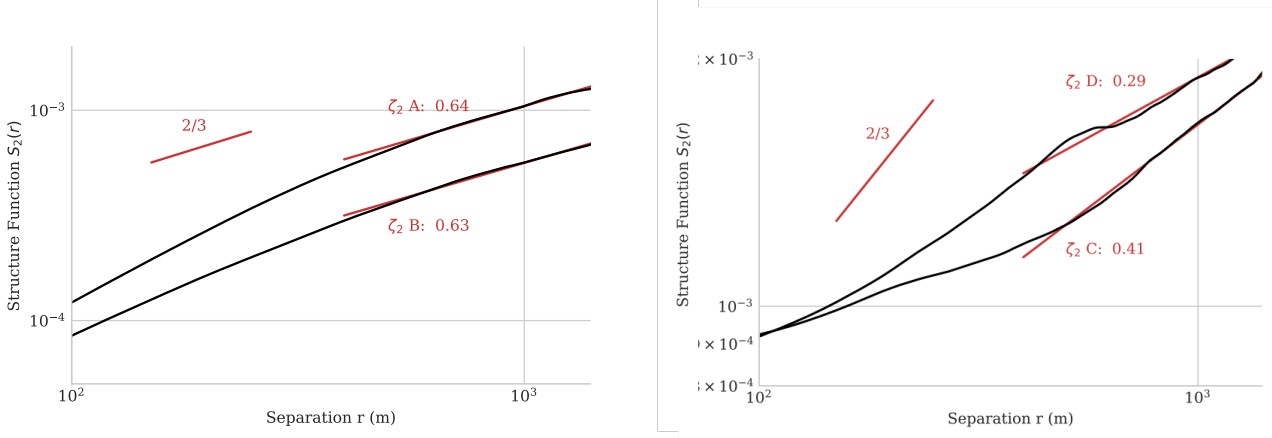

**Figure 8.** Structure functions for the maps in Figure 6. Left: Flightlines A and B. Right: Flightlines C and D.

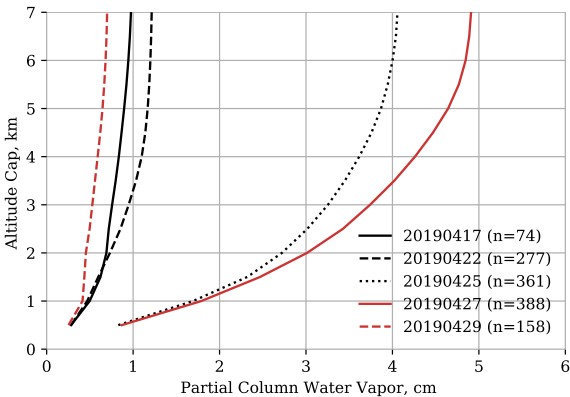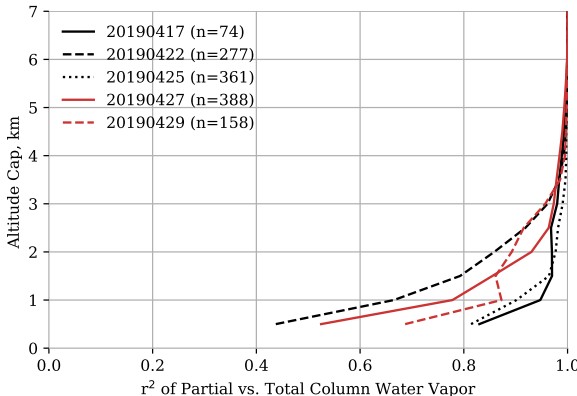

**Figure 9.** Data from HALO flights suggests that the PBL dominates the total column variability. Left: Partial column water vapor for five flights calculated at different capping altitudes. Right: Correlation coefficients between the lowest atmospheric layer and the total column water vapor for different capping altitudes.

for altitudes 4 km or lower, and 1 km vertical averaging for altitudes higher than 4 km. This along-track averaging provided an optimal balance of spatial resolution and sensitivity. We integrated the column water vapor vertically to obtain partial column water vapor (PCWV) and calculated the squared Pearson correlation coefficient ($r^2$) between the PCWV up to different altitudes, and that up to 7 km, which approximated the Total Column Water Vapor (TCWV). Picking a higher altitude would have severely restricted the number of valid lidar footprints due to the flight altitudes, and there was also very little vapor content above 7 km. To create a dataset which was most similar to the AVIRIS-NG flight data, we extracted HALO data over ocean in the least-cloudy parts of each flight, with valid water vapor retrievals from the lowest bin up to 7 km. Measurements occurred approximately every 200 m and were averaged over 3 km. We required at least 90 % of the 200 m footprints to be non-cloudy within a given 3 km along-track measurement, and then identified "least cloudy" larger areas by moving a 750 km window along the flight line and requiring >20 % of the 3 km spacing profiles be clear.

Figure 9 shows that in every flight, over 70% of the variance in TCWV is explained by altitudes less than 2 km, and more than 90% of the variance is explained by variance at altitudes less than 3 km. While the horizontal resolution and meteorology of HALO and AVIRIS-NG flights are different, this provides quantitative evidence that the $\zeta_2$ values derived here most likely refer to bulk variability within the PBL. This adds to mesoscale studies finding that processes in the PBL largely control the total atmospheric water vapor content (Couvreux et al., 2009). It also suggests that, as a component of a larger measurement and assimilation system, the total column measurement could provide probabilistic constraints on PBL water vapor.

## 6 Conclusion

This paper describes an approach for mapping column average atmospheric water vapor at sub-kilometer spatial scales with remote VSWIR imaging spectroscopy. We validate the method by comparison with in-situ AERONET observations. We then

map column water vapor using reflected solar sunglint over ocean surfaces in several flightlines from an airborne campaign where favorable solar angles permitted a uniquely high resolution measurement. We find scaling behavior broadly consistent with prior studies, with high repeatability across different observations of the same airmass. The experiment is limited in some respects; for example, it provides only a total-column measurement, making it less compatible with height-resolved structure functions (Kahn et al., 2011). Moreover, stringent requirements on observation geometry limited the dataset to a few representative flightlines. Nevertheless, it provides a proof of concept that VSWIR imaging spectroscopy, primarily used for measurements of surface phenomena, can also provide accurate water vapor maps for atmospheric studies. The key innovation afforded by VSWIR water vapor retrievals will be the spatial scale, accuracy, and retrieval capability over land surfaces - all of which improve on various aspects of MODIS bi-spectral or passive microwave imaging techniques.

Future measurement campaigns aiming to observe water vapor with this approach should consider both the solar geometry and signal to noise level, either of which could limit the resolution. In this study, we limited the influence of solar geometry by careful selection of observing conditions. When this is not possible, computer techniques might be used to remove the spatial blurring effect of low solar angles. The effective spatial response function could be calculated and deconvolved from the measured vapor field. It may also be possible to reduce the impact of solar zenith angles by measuring the structure function orthogonally to the solar direction, a premise that will be explored in future work. Regardless of whether such compensation is possible, the next generation of orbital VSWIR imaging spectrometers should significantly increase the data available for fine-scale mapping of atmospheric water vapor.

*Code and data availability.* Code used for the calculations described in this manuscript is available in the ISOFIT code repository under the Apache 2.0 open source license (https://github.com/isofit/isofit). The airborne datasets described in this paper are available from the Jet Propulsion Laboratory (https://aviris.jpl.nasa.gov for AVIRIS-C, and https://avirisng.jpl.nasa.gov for AVIRIS-NG). AERONET data is available from the AERONET web site, (https://aeronet.gsfc.nasa.gov/)

*Author contributions.* D. R. Thompson contributed to the writing, algorithm design, experimental concept and execution. P. G. Brodrick contributed to code development, algorithm design and implementation, interpretation of the experiments, and writing. R. O. Green planned, coordinated, and managed the AVIRIS-C and AVIRIS-NG campaigns. B. H. Kahn provided scientific motivation, analsyis of MERRA-2 data, and writing. M. D. Lebsock contributed to the measurement concept, data interpretation, and writing. M. Richardson contributed to the measurement concept, code evaluation and validation, analysis of HALO data, and writing.

*Competing interests.* The authors have no competing interests with the investigation described in this document.

*Acknowledgements.* We thank the AERONET station PI(s) and Co-I(s) and their staff for establishing and maintaining the sites used in this investigation. Some of the Cimel Sun-Photometer data was collected by the U.S. Department of Energy as part of the Atmospheric Radiation Measurement Program User Facility (ARM) and processed by the National Aeronautics and Space Administration's Aerosol Robotic Network (AERONET). We thank the members of the AVIRIS-NG team who participated in data acquisition and analysis, including

Michael Eastwood, Sven Geier, Mark Helmlinger, Winston Olson-Duvall, and Sarah Lundeen. AVIRIS-NG is sponsored by the National Aeronautics and Space Administration (NASA) Earth Science Division. We also thank Amin R. Nehrir and Brian Carroll who assisted with HALO data analysis. This research was carried out at the Jet Propulsion Laboratory, California Institute of Technology, under a contract with the National Aeronautics and Space Administration. We also acknowledge the NASA Earth Science Division's AVIRIS-NG instrument and the data analysis program "Utilization of Airborne Visible/Infrared Imaging Spectrometer Next Generation Data from an Airborne

Campaign in India" NNH16ZDA001N-AVRSNG.

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
