# Peer review of "Spectroscopic Imaging of Sub-Kilometer Spatial Structure in Lower Tropospheric Water Vapor"

_Atmospheric Measurement Techniques, 2020_

## Referee Comment (RC1) · Anonymous Referee #1 · 21 Sep 2020

General comments

This manuscript presents spatially highly resolved two-dimensional AVIRIS-NG spectrometer measurements of the tropical water vapor column over the Bay of Bengal. Focus of this case study is to quantify the water vapor variability using second-order structure functions. The paper is suitable for AMT as it presents an innovative methodical approach to quantify the variability. I have three general comments.

First, unfortunately, the presented data set is very small. It only covers four about 9x2km large rectangular domains on two different days. I understand that these represent the best available high-resolution measurements over water. But still, this is

disappointing. How nice would it be to have a complementing example over land, or over a land-sea interface, or in a different climate zone, etc., even if the data were a little noisier. Please explain why your data set is so small, or show more.

Second, these few data, displayed in Figures 5 to 7, lack more explanation and description. Figures 5 and 6 show irregular patterns of water vapor column variability that could be related to turbulent processes in the boundary layer or at its top, but additional data and explanations on the meteorological conditions are lacking. Do we see eddies in the boundary layer, undulations of its top height, or features above, or a mixture of all? How would you interpret this variability? What are the underlying physical processes? For illustration, even already a photo out of the aircraft window could help set the scene a little. But it would be good to show more useful auxiliary data.

Third, I am lacking some discussion on possible implications due to the issue that the vertical water vapor distribution may be complex. The missing vertical resolution may lead to a superposition in the column between different atmospheric regimes, such as convective and non-convective, or boundary-layer and free-troposphere, blurring the variability in a particular layer and making the results in Figure 7 quite questionable. It is clear that most of the column is from the lower troposphere, but you should be more precise on this issue. How likely is it that variability in the mid- to upper troposphere is superposed to the lower troposphere patterns? At least, you should use a tropical humidity profile to show how much contribution to the column comes from the different layers.

Specific comments

Line 2: "total atmospheric column wv" but the title says ". . . Lower Tropospheric WV", better make it more homogeneous. This issue is related to my third general comment above, that some discussion on the vertical wv distribution is missing.

Line 62, "at the finest scales, small sample sizes can increase uncertainty. . .": usually, spectra and structure functions provide lowest uncertainties at the small scales due to
high sampling. This sounds like a contradiction. Furthermore, I do not see a relation with the sentence before on one-dim versus two-dim observations. Please explain.

Line 117, "tens or hundreds of meters": this is much too small, you probably mean "tens or hundreds of km"?

Line 130, "the spatial footprint projected on the ground is not radially symmetric; it is long and thin": this may entail issues with spatially oriented wv structures, you may want to comment on this?

Table 1: an extra column giving the length of the flight line in km would be fine.

Figure 6: make a white box in Fig 5 to show the location of Fig 6.

Technical corrections

Line 252, "pathological effects", better "issues"

Caption of Fig 5: "A white arrow in B indicates. . ."

---

## Referee Comment (RC2) · David Adams (Referee) · 16 Dec 2020

Review of Thompson et al. 2020 submitted to AMT
by David K. Adams
dave.k.adams@gmail.com

Recommendation:  Minor Revisions

General Comments:
The authors present a study employing an airborne VSWIR imaging spectrometer to examine very high spatial resolution  (sub-kilometer scale) column water vapor.  The technique is presented with sufficient detail and the results argue for similar structure function scaling exponents that are observed over many decades of scale; a perhaps somewhat surprising result.  As a proof of concept, the manuscript is more than adequate for publication.  Perhaps one weakness would be the need to expand the literature a bit to include some examples of more meteorological/climatological applications focusing on column/precipitable water vapor spatial/temporal structure.  This would help better motivate the study and draw greater interest for a broader community.  This broader literature review wouldn´t need to be more than one paragraph.  One other point to address is to clarify some of the langauge.  It is confusing at times, for example, which spatial scales you are referring to.  See below in my minor comments.

Minor Comments:

Line 13.  Water vapor and cloud formation are important for all numerical models of the atmosphere, not just General Circulation Models (GCMs).  Even at very high resolutions where deep convection is resolved (i.e. ~ km-scale) such as "Cloud-Resolving models" or "Convection-Permitting Models"  and even  for "Large Eddy Simulations" (~100m-scale), cloud microphysical processes which critically depend on water vapor are still parameterized.

Line 19-20  You should be clear as to what spatial scales you're referring to.  Convective and non-Convective systems would typically be 10s of kms to maybe 100km and quasi-geostrophic motions would be 1000km and greater from Edwards et al., (2019).

Line 24  "but in general water vapor variability  is considered horizontally isotropic." This idea is a bit unclear, what exactly do you mean horizontally isotropic particularly with respect to spatial scales?

Line 35  "consistent with 2/3 over distances of multiple kilometers."  Do you mean several kilometers here?

 Line 41  "at scales above 11 km"  I assume you mean at scales greater than 11km.  " Above 11km" sounds as if you are speaking in the vertical sense.

Line 49  Just write  "...compared them to GCMs, ...."

 Line 58.  "These studies contribute to a growing body of literature on water vapor scaling."
I think it would be good to include a paragraph on some of these studies.  Not only techniques for measuring PWV, but theoretical as well as applied studies to meteorology/climate.  Are there modelling studies which have used these scaling arguments as metrics?  PWV is certainly a critical if not "the" critical variable for deep convection in the Tropics. There are numerous studies observational, modeling and theoretical which focus on this relationship, including temporal and spatial scaling arguments.  This would help motivate this study a bit more and why it has more "global" importance.

Line 85  Write  "... build upon these results."

Line 105  Write out RTM.  I assume you mean Radiative Transfer Model, but just to be clear for the reader.

Line 155 "We solve it with a trust region gradient descent optimization."  You might want to clarify what this is.

Line 183  I think it would be clearer to write  "leave-one-out cross-validation"

Line 203 Spell out "AFGL"

Line 213   Write  "Some discrepancies in the optical paths remain, which become larger for column water vapor in the free troposphere than in the planetary boundary layer."

Line 251  Write "This artifact, indicated by a white arrow,  may be related to pathological effects from the sun glint bidirectional reflectance distribution or the aircraft shadow.  Therefore, it was excluded from the statistics."

Line 258  Write "second-order"

Figures:

Figure 1. Left: ... with gray arrows. The arrows look red to me.

Figure 2. Left: Write ".... In reality,  the sun ..."

Figure 6. Left:  flightine is mispelled.

---

## Author Comment (AC1) · 13 Jan 2021

We thank the reviewer for their feedback and suggestions for additional discussion topics. We have revised the manuscript accounting for these recommendations. A point by point explanation of our changes follows.

[Figure]

**General Comments**

This manuscript presents spatially highly resolved two-dimensional AVIRIS-NG spectrometer measurements of the tropical water vapor column over the Bay of Bengal. Focus of this case study is to quantify the water vapor variability using second-order structure functions. The paper is suitable for AMT as it presents an innovative methodical approach to quantify the variability. I have three general comments.

First, unfortunately, the presented data set is very small. It only covers four about 9x2km large rectangular domains on two different days. I understand that these represent the best available high-resolution measurements over water. But still, this is disappointing. How nice would it be to have a complementing example over land, or over a land-sea interface, or in a different climate zone, etc., even if the data were a little noisier. Please explain why your data set is so small, or show more.

We agree and would also have preferred a larger dataset. In this case, the limiting factor was not the quantity of imaging spectrometer data, but rather the near-zenith solar angles needed for a high spatial resolution measurement. This condition only happens near or in the tropics, outside the range of most AVIRIS-NG campaigns. Limiting ourselves to solar zeniths less than 10 degrees reduced us to the flight days reported here. Now, having established the technique, we believe that we can apply it in the future to much larger datasets that will be made available by NASA's EMIT mission and the anticipated SBG investigation. Additional work in preparation will demonstrate algorithmic techniques to relieve the solar zenith requirement, which should help generalize it to a much larger range of latitudes. We have included new text in the discussion to make this explicit.

Second, these few data, displayed in Figures 5 to 7, lack more explanation and description. Figures 5 and 6 show irregular patterns of water vapor column variability that

could be related to turbulent processes in the boundary layer or at its top, but additional data and explanations on the meteorological conditions are lacking. Do we see eddies in the boundary layer, undulations of its top height, or features above, or a mixture of all? How would you interpret this variability? What are the underlying physical processes? For illustration, even already a photo out of the aircraft window could help set the scene a little. But it would be good to show more useful auxiliary data.

We agree, and have added four new graphics with contextual information based on MERRA-2 reanalysis of the overflight days (Figure 1 of this document). We have also updated the text with our interpretation. The atmospheric conditions were generally similar between the 12th and 14th, with light trade winds and the lack of an obvious inversion to stratify boundary layer processes. There were also some differences. Wind velocity changed slightly, but was not obviously tied to any change in atmospheric turbulence. The relative humidity was generally higher on the 14th. The lapse rate was slightly more variable: 8 K km$^{-1}$ at 760 hPa as opposed to 7 K km$^{-1}$ on the 12th. These changes were consistent with a slightly more turbulent atmosphere and a shallower scaling exponent, though there was no obvious step change in atmospheric stability.

Third, I am lacking some discussion on possible implications due to the issue that the vertical water vapor distribution may be complex. The missing vertical resolution may lead to a superposition in the column between different atmospheric regimes, such as convective and non-convective, or boundary-layer and free-troposphere, blurring the variability in a particular layer and making the results in Figure 7 quite questionable. It is clear that most of the column is from the lower troposphere, but you should be more precise on this issue. How likely is it that variability in the mid- to upper troposphere is superposed to the lower troposphere patterns? At least, you should use a tropical humidity profile to show how much contribution to the column comes from the different layers.

We agree this issue deserves a more nuanced treatment, and have introduced a new discussion section to address the issue of vertical sensitivity. There we analyze a large dataset of measured atmospheric water vapor profiles from an airborne lidar campaign (Figure 2 of this document). Specifically, we calculate the correlation coefficients between the total column water vapor content and that of the lowest atmospheric layers. A new figure quantifies the relationship. The strength of the correlation varies slightly for different flights. However, in all three cases the lower troposphere dominates the short-lengthscale variability in atmospheric water vapor. In all cases over 70% of the variance is explained by the lower two kilometers, and over 90% by the lower three kilometers. This suggests that the total column measurement an informative indicator of lower tropospheric variability. Naturally the relationship is statistical rather than direct, and we agree with the reviewer that it is important to include this explanation. With the reviewer's permission, we will borrow some of their phrasing directly for the new discussion.

**Specific comments**

Line 2: "total atmospheric column wv" but the title says ". . . Lower Tropospheric WV", better make it more homogeneous. This issue is related to my third general comment above, that some discussion on the vertical wv distribution is missing.

We agree and have changed the title to the "atmospheric water vapor," reserving the issue of vertical sensitivity for the new discussion section.

Line 62, "at the finest scales, small sample sizes can increase uncertainty. . .": usually, spectra and structure functions provide lowest uncertainties at the small scales due to

high sampling. This sounds like a contradiction. Furthermore, I do not see a relation with the sentence before on one-dim versus two-dim observations. Please explain.

Thank you for identifying this confusing explanation. Our point is that the structure function measurements fall into two categories: vast two dimensional maps by AIRS, with millions of datapoints but coarse resolution; and aircraft data, which have finer spatial resolution but also smaller sample sizes since they only acquire samples along a flight trajectory. We have adjusted the text to clarify.

Line 117, "tens or hundreds of meters": this is much too small, you probably mean "tens or hundreds of km"?

In fact the phrase was written as intended. The left panel of Figure 2 shows that, after accounting for the vertical distribution of water vapor, observations at low solar zenith angles do have effective spatial resolutions in this range; the coarsest resolution indicated on the vertical axis is 1000 m. Since only a small fraction of these have resolutions finer than 100 meters, we have changed the phrase to "hundreds of meters" to avoid confusion.

Line 130, "the spatial footprint projected on the ground is not radially symmetric; it is long and thin": this may entail issues with spatially oriented wv structures, you may want to comment on this?

We have added some additional text here. "For isotropic structure functions, the effective spatial resolution thus constitutes a worst case, with the true resolution improving as one calculates structure functions in directions more orthogonal to the sun." Future work will investigate this strategy as a method to measure accurate structure functions at larger solar zenith angles.

Table 1: an extra column giving the length of the flight line in km would be fine.

Done.

Figure 6: make a white box in Fig 5 to show the location of Fig 6.

Done.

**Technical corrections**

Line 252, "pathological effects", better "issues"

Done.

Caption of Fig 5: "A white arrow in B indicates. . ."

Done.

[Figure]

**Fig. 1.** MERRA-2 Reanalysis

[Figure]

[Figure]

**Fig. 2.** Airborne Lidar Profiles

---

## Author Comment (AC2) · 13 Jan 2021

We thank the reviewer for their feedback, which we have duly incorporated. A point by point explanation of our changes follows.

**General Comments**

The authors present a study employing an airborne VSWIR imaging spectrometer to examine very high spatial resolution (sub-kilometer scale) column water vapor. The

technique is presented with sufficient detail and the results argue for similar structure function scaling exponents that are observed over many decades of scale; a perhaps somewhat surprising result. As a proof of concept, the manuscript is more than adequate for publication. Perhaps one weakness would be the need to expand the literature a bit to include some examples of more meteorological/climatological applications focusing on column/precipitable water vapor spatial/temporal structure. This would help better motivate the study and draw greater interest for a broader community. This broader literature review wouldn Ìągt need to be more than one paragraph

We agree, and have added a new paragraph in the opening that provides context on meteorological and climatological implications of PWV variability: "The complex spatial distribution of atmospheric water vapor surrounding clouds and precipitation structures has important consequences for parameterizing moist processes in atmospheric models. At the scale of General Circulation Models (GCMs), water vapor plays an important role in tropical moist convection and its associated precipitation (Tompkins et al., 2001; Bretherton et al., 2004). The mean and variability of precipitation rate in the tropics are strongly dependent on the atmospheric water vapor (Peters et al., 2006; Holloway et al., 2010), a fact which has implications for parameterizing convection. Another ubiquitous property of convection is its tendency to aggregate (Bretherton et al., 2005). There is evidence the degree of aggregation will change as the climate warms, potentially changing the cloud feedback (Wing et al., 2019). Models (Muller et al., 2015) and observations (Lebsock et al., 2017) suggest that the tendency of convection to aggregate depends on the degree of spatial variance in the water vapor field. Over land surfaces with heterogeneous surface conditions the variability in atmospheric water vapor can be larger and is seen as a critical component of the timing of deep convection (Stirling et al., 2004; Wulfmeyer et al., 2006). These variations in water vapor over convective continental environments are primarily driven by variability below 2 km and within the Planetary Boundary Layer (PBL) (Couvreux et al., 2009). Accurate water vapor parameterization is also important for Cloud-Resolving or Convection-Permitting

models operating at kilometer scales, and Large Eddy Simulations at sub-kilometer resolution. Across all scales, water vapor variability, and its coupling to cloud types and multi-scale organization, is key for advancing the parameterization and simulation of cloud processes."

One other point to address is to clarify some of the language. It is confusing at times, for example, which spatial scales you are referring to. See below in my minor comments.

(addressed below)

**Minor comments**

Line 13. Water vapor and cloud formation are important for all numerical models of the atmosphere, not just General Circulation Models (GCMs). Even at very high resolutions where deep convection is resolved (i.e.     km-scale) such as "Cloud-Resolving models" or "Convection-Permitting Models" and even for "Large Eddy Simulations" ( 100m-scale), cloud microphysical processes which critically depend on water vapor are still parameterized.

We have added additional text to this effect: "Atmospheric water-vapor is highly variable in space and time. Clouds and precipitation structures are embedded within complex spatial distributions of water vapor that have important consequences for parameterization of moist processes in General Circulation Models (GCMs). In global models, the dynamic relationship between water vapor sources, sinks, and atmospheric mixing leads to highly variable humidity at the sub-grid scale. Accurate water vapor parameterization is also important for Cloud-Resolving or Convection-Permitting models operating at kilometer scales, and Large Eddy Simulations at sub-kilometer resolution. Understanding this variability, and how it couples to a myriad of cloud types and multi-scale organization, is key for advancing the parameterization and simulating cloud processes at all scales."

Line 19-20 You should be clear as to what spatial scales you're referring to. Convective and non- Convective systems would typically be 10s of kms to maybe 100km and quasi-geostrophic motions would be 1000km and greater from Edwards et al., (2019).

We now state explicitly that prior studies have applied structure functions to atmospheric phenomena at scales from tens to hundreds of kilometers.

Line 24 "but in general water vapor variability is considered horizontally isotropic." This idea is a bit unclear, what exactly do you mean horizontally isotropic particularly with respect to spatial scales?

Here we simply mean that the scaling does not have any preferential orientation, i.e. the structure function is rotationally symmetric. We have modified the text to make this clear.

Line 35 "consistent with 2/3 over distances of multiple kilometers." Do you mean several kilometers here?

Changed.

Line 41 "at scales above 11 km" I assume you mean at scales greater than 11km. " Above 11km" sounds as if you are speaking in the vertical sense.

Changed.

Line 49 Just write "...compared them to GCMs, ...."

Changed.

Line 58. "These studies contribute to a growing body of literature on water vapor scaling." I think it would be good to include a paragraph on some of these studies. Not only techniques for measuring PWV, but theoretical as well as applied studies to meteorology/climate. Are there modelling studies which have used these scaling arguments as metrics? PWV is certainly a critical if not "the" critical variable for deep convection in the Tropics. There are numerous studies observational, modeling and theoretical which focus on this relationship, including temporal and spatial scaling arguments. This would help motivate this study a bit more and why it has more "global" importance.

We have provided additional information and references in the introduction, described under main question 1 above.

Line 85 Write "... build upon these results."

Changed.

Line 105 Write out RTM. I assume you mean Radiative Transfer Model, but just to be clear for the reader.

Changed.

Line 155 "We solve it with a trust region gradient descent optimization." You might want to clarify what this is.

We have added a short explanation and a reference to the original text.

Line 183 I think it would be clearer to write "leave-one-out cross-validation"

Changed.

Line 203 Spell out "AFGL"

Changed.

Line 213 Write "Some discrepancies in the optical paths remain, which become larger for column water vapor in the free troposphere than in the planetary boundary layer."

Changed.

Line 251 Write "This artifact, indicated by a white arrow, may be related to pathological effects from the sun glint bidirectional reflectance distribution or the aircraft shadow. Therefore, it was excluded from the statistics."
Changed, incorporating additional suggestions by reviewer 1.

Line 258 Write "second-order"

Changed.

**Figures**

Figure 1. Left: ... with gray arrows. The arrows look red to me.

Changed.

Figure 2. Left: Write ".... In reality, the sun

Changed.

Figure 6. Left: flightine is mispelled.

Changed.

---

## Author Response (AR2)

**Spectroscopic Imaging of Sub-Kilometer Spatial Structure in Atmospheric Water Vapor: Response to Second Review Round**

David R. Thompson[1], Brian H. Kahn[1], Philip G. Brodrick[1], Matthew D. Lebsock[1], Mark Richardson[1], and Robert O. Green[1]

[1]Jet Propulsion Laboratory, California Institute of Technology, Pasadena, CA 91103 USA

**Correspondence:** David R. Thompson (david.r.thompson@jpl.nasa.gov)

*Copyright statement.* Copyright 2021 California Institute of Technology. Government Support Acknowledged.

We thank reviewer 1 for their feedback, which we have duly incorporated. A point by point explanation of our changes follows.

The authors have made substantial improvements to the manuscript and added valuable figures. I am satisfied with the revised version and have only the following minor points: a) Title: I leave it up to the authors whether they would want to change the title slightly, following my recommendation, or not.

Thank you. We have decided to use the current version of the title.

b) Line 211: horizontal

We have modified this sentence to indicate that we dilate masks in the horizontal dimension.

c) Fig 4, rel humidity: x-axis is not in percent

We have modified the image so that the horizontal axis is labeled correctly. Thank you for this catch.

d) Line 484: the authors may use my phrasing for their new discussion.

Thank you.